# Vapor Phase Modification for Selective Enrichment of Grafted Styrene/Acrylonitrile onto Carbon Nanotubes Via ATRP

Maryam Azadbakht [1] , Elnaz Esmizadeh [1,2,*] , Ali Vahidifar [1], Tizazu H. Mekonnen [2] and Mehdi Salami-Kalajahi [3]

1    Faculty of Engineering, Department of Polymer Science and Engineering, University of Bonab, Bonab 5551761167, East Azerbaijan, Iran; azad.1989.m@gmail.com (M.A.); vahidifar.ali@gmail.com (A.V.)
2    Department of Chemical Engineering, Institute of Polymer Research, University of Waterloo, Waterloo, ON N2L3E9, Canada; tizazu.mekonnen@uwaterloo.ca
3    Faculty of Polymer Engineering, Sahand University of Technology, P.O. Box 51335-1996, Tabriz 51335/1996, East Azerbaijan, Iran; m.salami@sut.ac.ir
*    Correspondence: Elnaz.Esmizadeh@uwaterloo.ca

**Abstract:** Nitric acid vapor phase oxidation of multi-walled carbon nanotubes (MWCNTs) was proposed as a promising technique to fabricate poly styrene-co-acrylonitrile (SAN)-grafted-CNTs via atom transfer radical polymerization (ATRP). The in-situ ATRP grafting approach was successfully employed to graft polystyrene (PS), SAN and polyacrylonitrile (PAN), onto the convex surfaces of pristine MWCNTs (PCNT) and acid-functionalized MWCNTs (FCNT). Fourier transform infrared spectroscopy (FTIR), proton nuclear magnetic resonance ($^1$H-NMR), and thermogravimetric analysis (TGA) confirmed the effectiveness of the modification via the ATRP grafting approach. The molar composition of acrylonitrile in the synthesized copolymer on the surface of CNTs for an FCNTs was calculated to be about 80% and 67.5% by $^1$H-NMR and TGA respectively, whereas the value is lower for PCNTs. Morphological studies showed that SAN-grafted FCNTs exhibit rougher surface morphology compared to the SAN-grafted PCNTs. Moreover, the higher diameter of the FCNTs indicated the higher polymer content, which was coated onto CNTs functionalized by vapor-phase oxidation. Therefore, the vapor phase oxidation strategy employed in this study could be utilized as a general method to prepare CNTs which can serve as an ATRP macroinitiator for the fabrication of various polymer grafted CNTs.

**Keywords:** atom transfer radical polymerization (ATRP); SAN-grafted-MWCNT; CNTs; CNT oxidation; grafting

## 1. Introduction

Carbon nanotubes (CNTs) are categorized among the most impressive recent discoveries in chemistry and materials science [1,2]. CNTs have raised much enthusiasm during the latest years due to their intrinsic extraordinary electrical and mechanical properties [3]. However, transferring their properties into bulk and real-life materials is restricted by their insolubility in most solvents [4,5] and their low dispersibility in polymer matrices [6,7]. The functionalization and surface modification of CNTs are widely accepted and promising measures to improve their dispersion in solvents and enhance their compatibility with polymers by tailoring their structure and surface properties, favoring the fabrication of CNT-based composites [8]. Depending on the method of modification strategy, oxygen-containing groups, mostly carboxyl and hydroxyl, are formed on the primary CNTs graphitic surface [9,10]. Such oxygen-containing groups can help to diversify further tailoring of CNTs and change their performance in a wide range of materials [9,10].

Higher oxygen content in multi-walled carbon nanotubes (MWCNTs) is indicative of the presence of defects in the form of additional functional groups (hydroxyl, carbonyl, or carboxyl) [11,12]. Following the defect-producing oxidation process on the primary

surface, further formation of defects can be achieved in the intermediate and final products. The presence of defects on the outer surface of CNTs not only impacts the morphological stability of oxidized material but also determines their electrical and other functional properties [11,12]. The oxidation of CNTs by wet chemical methods was reported by Kovtyukhova et al. [11–13]. One of the interesting applications of CNTs is to fabricate polymer-based composites containing conductive carbonaceous materials. This category has been the most attractive nominee to meet urgent requirements such as chemical stability, lightweight, design flexibility, and cost-effectiveness over conventional metal-based materials [14,15]. Grafting macromolecules onto the rounded walls of CNTs, with the intent of improving their processability and extending their applications, has been explored for several years [16]. Recently, the grafting-from approach has been employed to functionalize MWCNTs by the bulk in-situ grafting of polymer chains onto the convex walls of CNTs [17]. Many linear polymers including polystyrene (PS) [17], poly (sodium 4-stryrenesulfonate) [18], poly (methyl methacrylate) [19], polyimide [20,21], poly (2-vinylpyridine) [22], poly(ethylene glycol) [23], poly (vinyl alcohol) and its related copolymer poly(vinyl acetate-co-vinyl alcohol) [24], and poly (m-aminobenzenesulfonic acid) [25] have been successfully attached onto CNTs.

Although substantial studies have been accomplished on the grafting of polymers onto CNTs, to the best of our knowledge, there is no published data on the grafting of styrene-co-acrylonitrile (SAN) onto the surface of CNTs in the scientific literature. SAN holds considerable importance in diverse and demanding application areas, such as automotive, packaging, electronics, and medical applications [26,27]. The successful grafting of styrene-co-acrylonitrile onto the surface of CNTs can open up a path to the invention of functional nanomaterials and polymer composites with a fine dispersion of functionalized CNTs, which can be helpful in electronic and a range of other applications. Although traditional radical polymerization techniques are suitable to increase homopolymers or random copolymers under mild circumstances, controlled radical methods are more appealing in achieving beneficial features and architectures with steady molecular weights and low polydispersities [26,27]. The atom transfer radical polymerization (ATRP) has been used widely to functionalize CNTs through the progress of polymer brushes on the surface of nanotubes [28,29].

The knowledge of how to control polymer content and selective polymer composition on the surface of CNTs is of fundamental importance in order to determine the final properties, performance, and application of the CNT-polymer hybrid composites. Thus, the objective of this work was to investigate the effect of pre-functionalization on the productivity of ATRP on CNTs. In this study, the ATRP method was employed for the grafting of SAN copolymer on the convex surface of MWCNT. The effect of vapor phase functionalization of MWCNTs via nitric acid prior to ATRP was also investigated on the level of grafting and material characteristics of the product. The SAN-grafted-pristine MWCNTs (SAN-g-PCNT) and SAN-grafted-acid functionalized MWCNTs (SAN-g-FCNT) were explored via Fourier transform infrared spectroscopy (FTIR), proton nuclear magnetic resonance ([1]H-NMR), and thermal analyses, including a differential scanning calorimetry (DSC) and thermogravimetric analysis (TGA) methods.

## 2. Experimental

### 2.1. Materials

Pristine MWCNTs (PCNTs) with a diameter range of 20–30 nm and a length range of 10–30 μm were supplied from Neutrino Corporation (Tehran, Iran). Other reagents and materials used for the modification of MWCNTs in this work were styrene (Sigma-Aldrich, 98%, St. Louis, MO, USA), Acrylonitrile (Merck, 98%, Darmstadt, Germany), Copper(I) bromide (Cu(I)Br; Sigma-Aldrich, 98%, Taufkirchen, Germany), ethyl α-bromoisobutyrate (EBiB; Sigma-Aldrich, 97%, Darmstadt, Germany), N,N,N′,N″,N″-pentamethyldiethylenetriamine (PMDETA; Sigma-Aldrich, 99%, Milwaukee, WI, USA), tetrahydrofuran (THF; Merck, 99%,

Darmstadt, Germany), N,N-dimethylformamide (DMF) (Sigma-Aldrich, 99%, Shanghai, China), and HNO$_3$ (Sigma-Aldrich, 69%, St. Louis, MO, USA).

### *2.2. Methods*

### 2.2.1. Vapor Phase Functionalization of CNTs

The functionalization of CNT was performed by a nitric acid (HNO$_3$) vapor phase functionalization as per the method reported by Xia et al. [30]. The experimental setup used here is illustrated schematically in Figure 1. The round bottom flask was first filled with 150 mL of 69% HNO$_3$ solution and heated under magnetic stirring to the boiling point (125 °C) in an ethylene glycol bath. 300 mg of PCNTs was then loaded into the reactor and heated up to 135 °C through a resistance heating coil controlled by a temperature controller. The functionalization reaction was then carried out for 2 h under the specific condenser design that completely prevents the reflux of liquid HNO$_3$ to the sample. This ensured that the functionalization process took place under a full vapor phase treatment avoiding the wetting of CNTs with the liquid acid. The condenser was connected to a round bottom flask that provided a recirculation system for continually recycling the solution. At the end of acid treatment, the heating of the ethylene glycol bath was turned off, and the reactor heating was maintained at 110 °C for another 2 h, allowing the CNTs to dry. Then, the acid-functionalized CNTs (FCNTs) were discharged from the reactor for further characterizations.

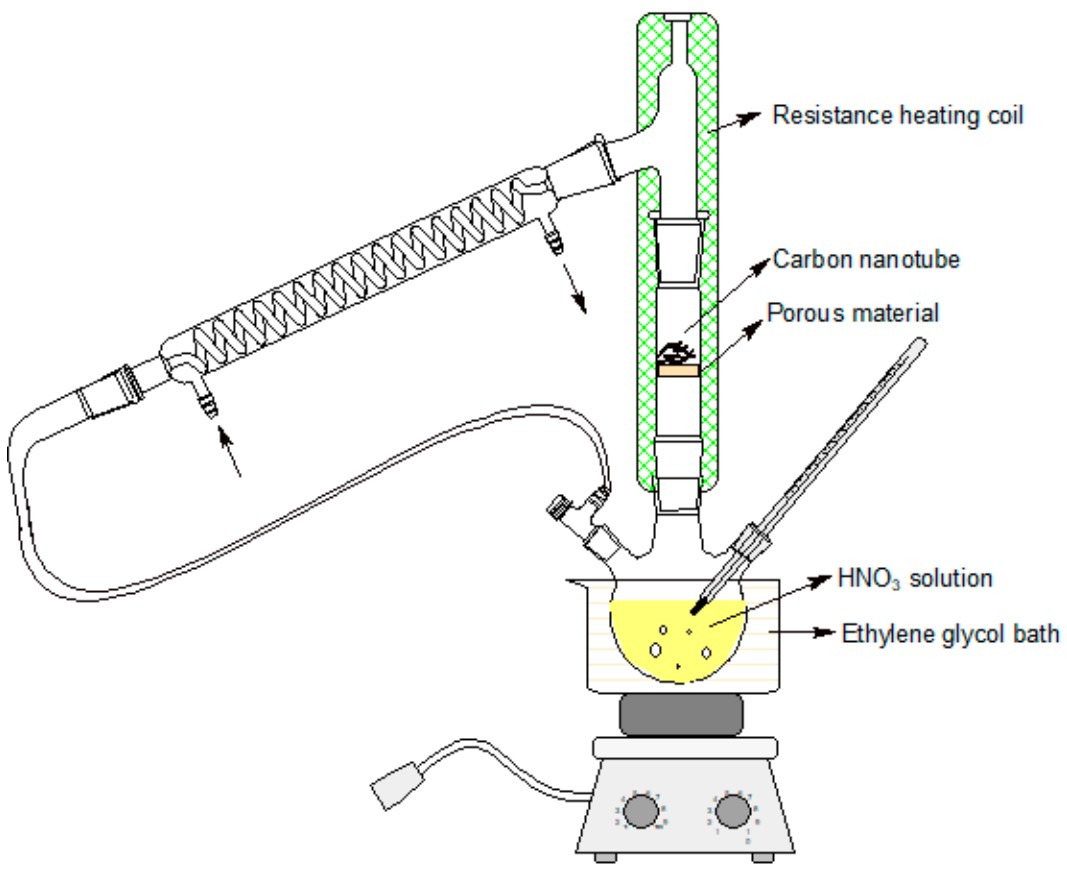

**Figure 1.** Schematic illustration of the set-up used for vapor phase functionalization of carbon nanotubes (CNTs) with HNO$_3$.

### 2.2.2. SAN-Grafted CNTs Via Atom Transfer Radical Polymerization (ATRP)

Copolymerization of styrene and acrylonitrile with a molar feed composition of 30/70 and PMDETA:CuBr:EBiB ratio of 1:1:1 was carried out in the presence of PCNTs and FCNTs. For this, 0.83 mL of PMDETA, 0.57g of CuBr, and 2.5 mL of DMF were introduced into a glass reactor flask with a lid and mixed for 15 min. 7 mL (0.0610 mol) of styrene, 0.6 mL

(0.0040 mol) of EBiB, and 3 mL (0.045 mol) of acrylonitrile were then added one by one into the glass flask and stirred under nitrogen and controlled temperature of 90 °C. The polymerization was allowed to continue for 72 h and was stopped by opening the flask.

### 2.2.3. Characterizations

Boehm titration analysis was performed to quantitatively evaluate the amount of carboxylic functional groups on FCNT samples after modification. For this, 50 mg of sample was added into a closed conical flask, which contains 200 mL of 0.05 N NaOH, and stirred at ambient temperature for 24 h to allow the solid material to equilibrate with the NaOH solution. Then, the 0.2 μL of filtered NaOH solution was titrated with 0.05 N HCl to determine the excess NaOH in the solution after the acid-base reaction with functionalized FCNT. The concentration of the acidic groups on MWCNT (mmole/g) was then adjusted to the amount of the reacted NaOH.

Thermogravimetric analysis (TGA) and derivative thermogravimetry (DTG) analysis was carried out on a Perkin–Elmer Pyris instrument (Norwalk, CT, USA) under a nitrogen atmosphere, using a ramp rate of 20 °C/min. The morphology of MWCNTs was analyzed using a Philips EM208 transmission electron microscope (TEM) under an accelerated voltage of 100 kV before and after treatment. To study the changes brought by vapor phase functionalization into the crystalline arrangement of MWCNTs, Raman spectra were collected using a Micro-Raman system (RM 1000 RENISHAW). A 50 mW laser operating at 785 nm was used as an excitation source, which was equipped with a Leica DMLM microscope and a Peltier-cooled charge-coupled device detector. Samples dispersed in deuterated chloroform were used to calculate the molar composition of styrene and acrylonitrile in the synthesized copolymer on the surface CNTs using proton nuclear magnetic resonance ($^1$H-NMR) (Ascend 600 MHZ, Bruker, Germany). The Fourier transform infrared spectroscopy (FTIR) spectra changes were evaluated on an Interspec 200-X FTIR spectrometer with 1 cm$^{-1}$ resolution, in the wavenumber range of 4000–400 cm$^{-1}$.

To study the glass-rubber transition temperature ($T_g$) of the polymer grafted onto MWCNTs, differential scanning calorimetry (DSC) was carried out using a 6000 Diamond model Perkin Elmer calorimeter (Perkin Elmer, Norwalk, CT, USA). For the DSC analysis, a sample of about 10 mg SAN-grafted sample was heated at a rate of 10 °C/min in the temperature range of 25 to 250 °C. The morphology change of SAN-grafted CNTs was investigated using scanning electron microscopy (SEM), the Hitachi S-3700N model (Hitachi, Tokyo, Japan), and the information including the nanotube diameter and its distribution were calculated manually with image J software.

### 3. Results and Discussion

The ATRP of styrene and acrylonitrile monomers on the surface of CNTs was performed in THF at 90 °C with CuBr as a catalyst and EBiB-MWCNTs as an initiator. The halogen-exchange technique of CuBr catalyst with boromo initiator was employed. The initiator and the monomers produced a tertiary alkyl radical and a secondary radical respectively, which provided improved control of the copolymerization. A schematic representation of the surface functionalization and the employed ATRP method for the grafting of SAN copolymers is presented in Figure 2.

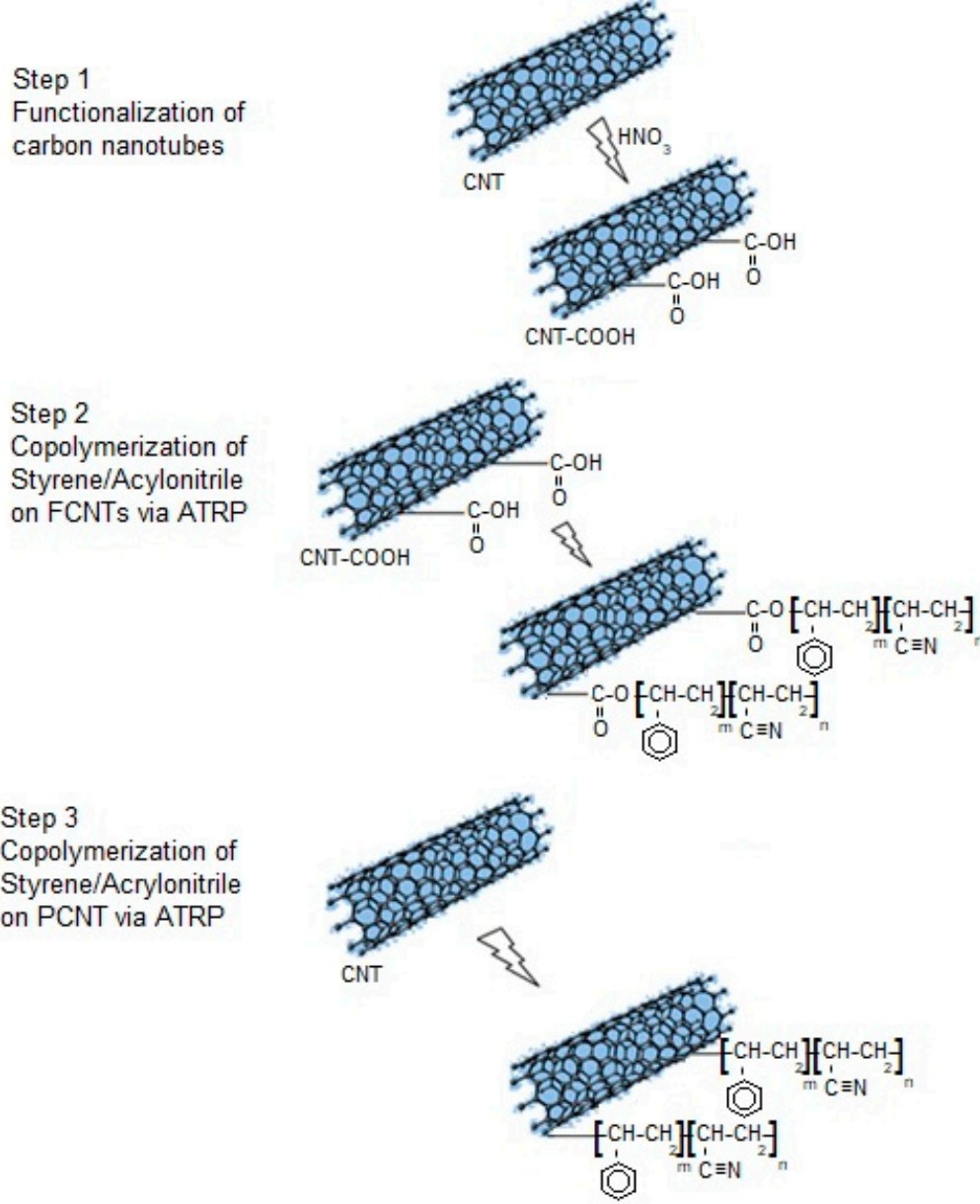

**Figure 2.** Schematic representation of functionalization CNT and synthesis of pristine CNT (PCNT) and acid-functionalized (FCNT).

Řesults of TEM analyses conducted to study the morphological changes that occurred on the PCNTs as a result of vapor phase functionalization are presented in Figure 3a,b. The PCNT (Figure 3a) exhibited long entangled tubes that consist mainly of smooth graphene layers, while modified ones, FCNTs (Figure 3b), illustrated shorter tubes with a crooked shape. Such nonalignment is rooted in the moderate extent of point defects upon oxidation, which degraded the smooth graphene layers present in PCNTs. This correspondingly caused a significant decrease in the graphitic structure on the external surface of the CNTs [31]. The surface of FCNTs is covered with densely packed clumps, attributed to the formation of defects associated with acid-functionalization. The introduction of hydrophilic functional groups into the surface of CNTs via surface oxidation could be helpful to avoid coagulation of CNTs. The presence of acid groups on the surface of CNTs could also enhance the polymer grafting reaction, due to the local increase in the reaction components on the surface of CNTs [5].

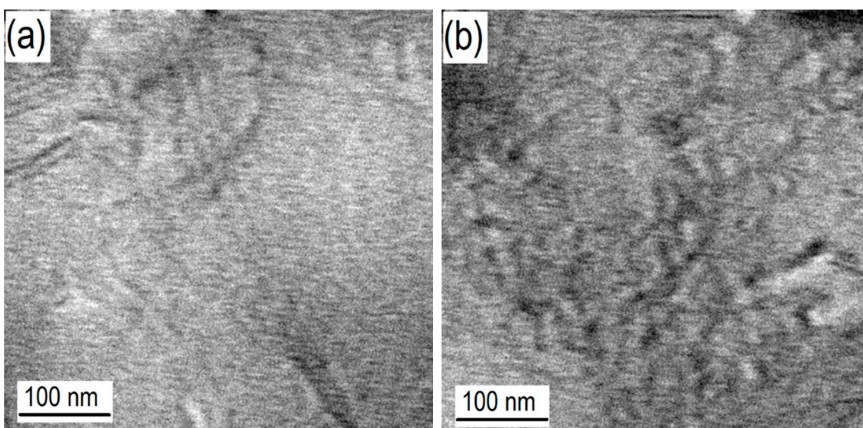

**Figure 3.** Transmission electron microscopy (TEM) micrographs of (**a**) PCNT and (**b**) FCNT after vapor phase treatment.

The details of the observed defects and changes in the arrangement of the CNT graphitic network are reflected on the Raman spectra features originating from carbon sp$^2$ vibration modes given in Figure 4. Thus, studying the evolution of the spectral profiles by Raman spectroscopy allows inferring information relative to the structural crystallinity/disorder degree and monitoring effects of the covalent modification of carbon bonds. The D-peak at 1310 cm$^{-1}$ indicated the impurities, lattice disorder, and defects in the CNTs' sidewall structure, while the G-peak at 1580 cm$^{-1}$, which originated from the in-plane vibrations of the graphene sheet, illustrated the crystalline graphitic carbons [32]. The relative intensity of D and G bands ($\frac{I_G}{I_D}$) was employed as a measure of the degree of graphitization for estimating the average domain sizes with graphitic order, i.e., crystallites ($L_c = 560\frac{I_G}{I_D}.E_L^{-4}$), where $E_L$ is the laser visible excitation energy (1.58 eV) [33]. The spectra confirmed that the two samples exhibit a slight difference in terms of defect density. Slightly higher defects were found in CNTs after vapor phase treatment, i.e., lower $\frac{I_G}{I_D}$ and $L_c$ values were obtained in FCNTs (inserted table in Figure 4). The reduction in graphitization index indicated the increase in the extent of covalent modification of C bonds produced by the formation of oxygenated groups through vapor phase functionalization.

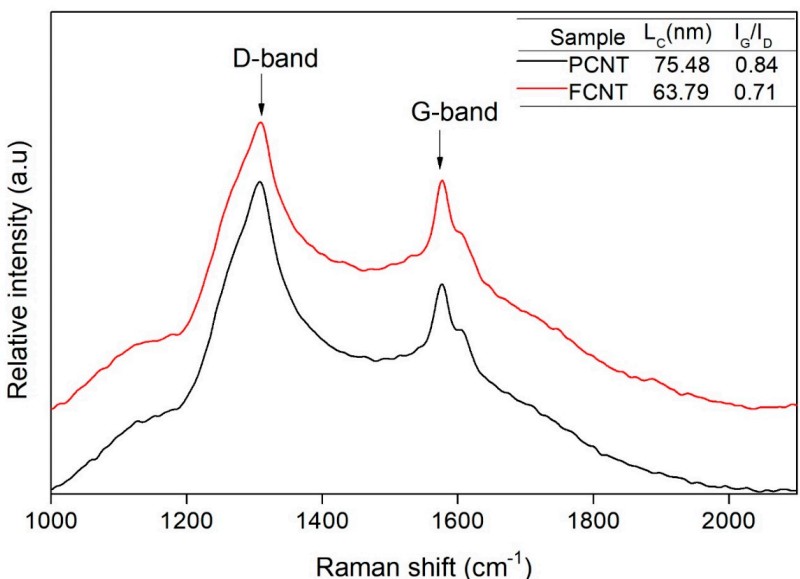

**Figure 4.** Raman spectra of pristine and vapor phase functionalized CNTs.

The concentration of carboxylic groups on PCNTs and FCNTs evaluated by Bohem titration was found to be 1.6 and 3.7 mmole/g, respectively. These values confirmed that vapor phase functionalization was achieved successfully, which was consistent with the Raman and TEM results. In addition, the concentrations of the COOH groups on the FCNTs can be reversely proportional to their residual weight prior to CNT degradation temperature (600 °C<) [32] since most of the organic functional groups are thermally decomposed below 600 °C under an inert atmosphere. The weight loss and the derivative weight loss curves associated with PCNTs and FCNTs are shown in Figure 5. The weight loss peak that occurred at a temperature range of 150–350 °C was related to the decomposition of a carboxylic group [34], while the next broad thermal degradation peak in the range between 350 and 600 °C could be associated with the degradation of anhydrides and lactones [35]. The mass loss values at 600 °C [32], which are mainly attributed to the degraded COOH, are 1.5 wt% and 16 wt% corresponding to 1.6 and 3.7 mmole/g of carboxylic groups respectively, that was obtained from the Boehm titration result.

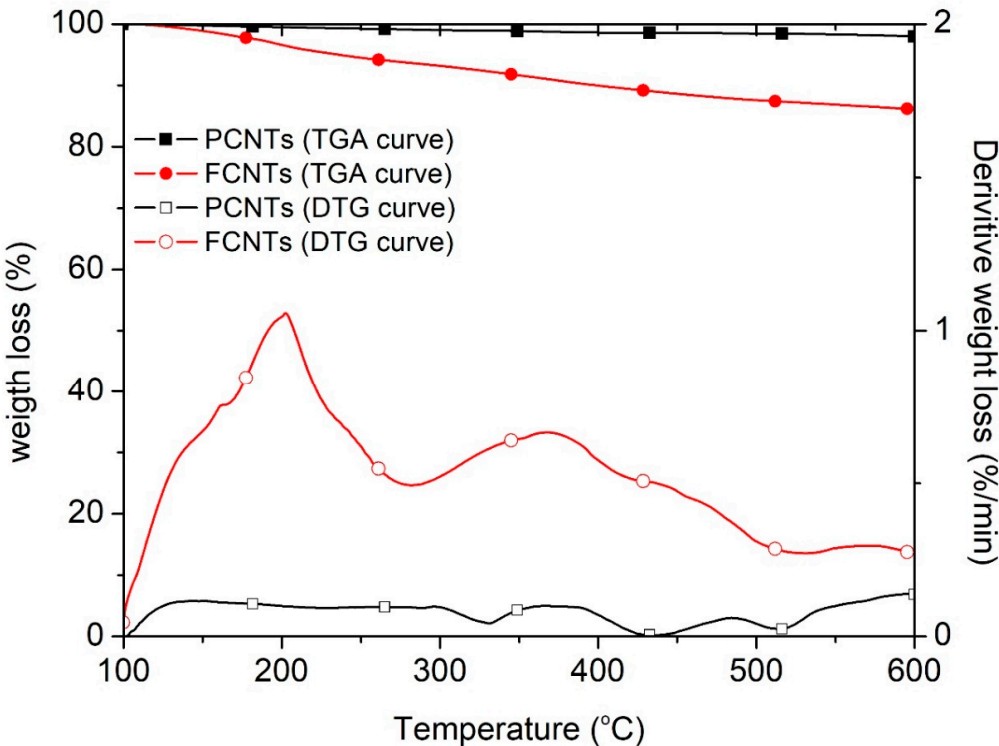

**Figure 5.** TGA/DTG thermographs of pristine and functionalized MWCNT.

FTIR spectra of SAN-g-PCNTs and SAN-g-FCNTs (Figure 6) were recorded and compared to confirm the existence of the synthesized SAN copolymer on the surface of CNTs. It can be observed from Figure 6 that both SAN-g-PCNTs and SAN-g-FCNTs spectra have peaks at 1600 and 1650 cm$^{-1}$, which were related to the C=C bond of the aromatic ring of styrene and the C=N bond of acrylonitrile cyanide group, respectively [36]. The peaks at 3436 and 1700 cm$^{-1}$ in SAN-g-FCNTs are associated with the presence of the hydroxyl group (OH) and the C=O stretching of the carboxylic acid group (COOH), respectively [37,38]. SAN-g-PCNTs spectra illustrate a strong peak at 2850 cm$^{-1}$, which corresponds to a commonly expected -C-H bond of CNT before functionalization. A remarkable reduction in the intensity of the –C–H peak at SAN–g–FCNT confirms the formation of new COOH bonds on the walls of FCNTs by functionalization which aids the involvement of FCNTs for the synthesis and formation of SAN on its surface.

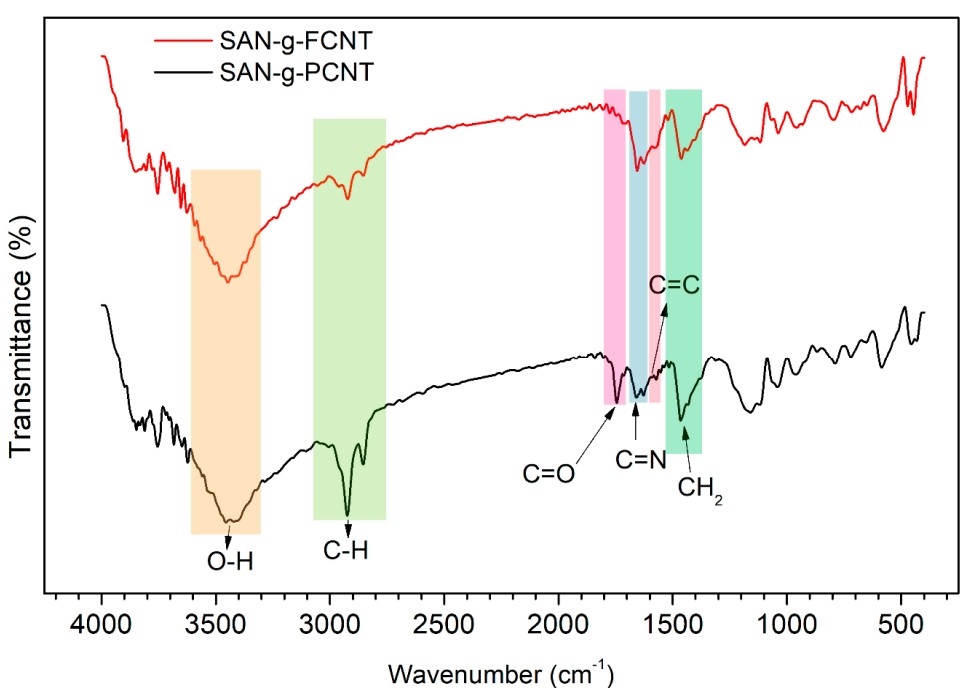

**Figure 6.** FTIR spectra for the SAN-grafted CNTs: SAN-g-PCNT and SAN-g-FCNT.

The covalent bonding of SAN onto MWCNTs both in the pristine and acid-functionalized state is further supported by [1]H-NMR. Figure 7 displays the [1]H-NMR spectroscopy results of the SAN-g-PCNT and SAN-g-FCNTs copolymer in deuterated chloroform after purification. The [1]H-NMR spectra of each material showed all the expected chemical bands, and the SAN grafting polymerization was also confirmed from the broad peaks at 1.3–1.8 ppm (f), at 2.1–2.6 ppm (e), and at 6.8–8.5 ppm (a, b), which correspond to the methylene protons of the styrene and acrylonitrile units, methine protons from styrene and acrylonitrile units, and the aromatic protons of styrene, respectively [39,40].

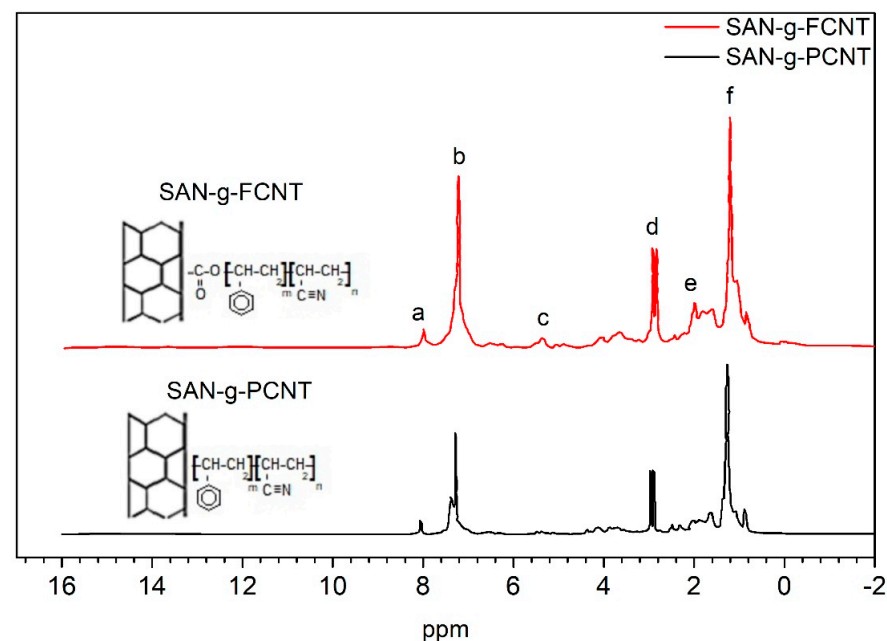

**Figure 7.** [1]H-NMR spectra for the SAN-grafted CNTs: SAN-g-PCNT and SAN-g-FCNT.

The C≡N group of acrylonitrile at 2.9–3.2 ppm (d) is a reaction junction copolymer (styrene/acrylonitrile) on MWCNTs [41]. The COOH and OH groups directly attached to the surface of MWCNTs after functionalization was confirmed by the peak in chemical shift of 4.5 to 5.5 ppm [42]. The copolymer composition was calculated from the $^1$H-NMR spectra using the integrals of peaks due to the aromatic peaks and the backbone protons of styrene and acrylonitrile (methylene and methine protons). The molar composition of acrylonitrile in the synthesized copolymer on the surface CNTs for FCNTs was calculated by about 80%, whereas the value is 78% for SAN-g-PCNTs. Thus, the results revealed that the synthesis of SAN copolymer on the surface of acid-modified CNTs yielded a SAN copolymer that has a higher molar composition of acrylonitrile. After vapor phase treatment of PCNTs with nitric acid, the MWCNTs were functionalized with carboxylic acid at the open ends and the defects of the side walls. The carboxylic acid groups can effectively involve the fixation of the ATRP initiator onto the MWCNTs' walls and the polymerization. Improved grafting of initiator groups onto the surface of FCNTs was reported by Shanmugharaj et al. [43].

Figure 8 shows the TGA, DTG, and DSC traces of SAN-g-PCNTs and SAN-g-FCNTs, which were used to determine the amount of polymer grafted onto MWCNTs and characterize the thermal behavior of SAN-g-PCNTs and SAN-g-FCNTs. The TGA thermograms (Figure 8a) of SAN-g-FCNT and SAN-g-PCNT exhibited two-step weight loss, in the range of 150–300 and 350–550 °C, respectively. The decomposition temperatures $T_{First}$ and $T_{Second}$, correlated to the first and second step of degradation, and the corresponding weight losses obtained from the TGA/DTG curves, were tabulated and presented in Table 1. While a first-step weight loss was 37.2% and 38.1%, the second step weight loss of 17.9% and 20.3% was detected for SAN-g-FCNT and SAN-g-PCNT respectively, as reported in Table 1 [44,45]. The first stage of degradation is related to the degradation of PAN and the second stage corresponds to the degradation of PS in SAN copolymer grafted onto MWCNTs [46,47]. Overall, the TGA data revealed an overall weight loss of 55.1% and 58.4% of SAN-g-FCNT and SAN-g-PCNT at 600 °C, respectively. Thus, SAN-g-FCNT showed higher polymer content, which was a consequence of surface modification of multiwall carbon nanotubes by nitric acid. The acrylonitrile content of SAN copolymer grafted on MWCNTs was calculated to be 67.5% and 65.2% in SAN-g-PCNT and SAN-g-FCNT. Although the SAN polymer content on the CNTs is not the same amount as the value estimated by $^1$H-NMR, the higher amount of polymer on the FCNT was in agreement with both the $^1$H-NMR and TGA analysis observations. This showed that FCNTs could be efficiently and selectively enriched with acrylonitrile, which can be identified with the high acrylonitrile/styrene ratio achieved in that case.

**Table 1.** Thermal results of SAN-g-FCNT and SAN-g-PCNT obtained from TGA and DSC.

| Sample | $T_{first}$ (°C) | wt.% Loss at 1st Step | $T_{second}$ (°C) | wt.% Loss at 2nd Step | $T_g$ (°C) |
|---|---|---|---|---|---|
| **SAN-g-PCNT** | 249 °C | 37.2 | 408 °C | 17.9 | 102 °C |
| **SAN-g-FCNT** | 266 °C | 38.1 | 410 °C | 20.3 | 105 °C |

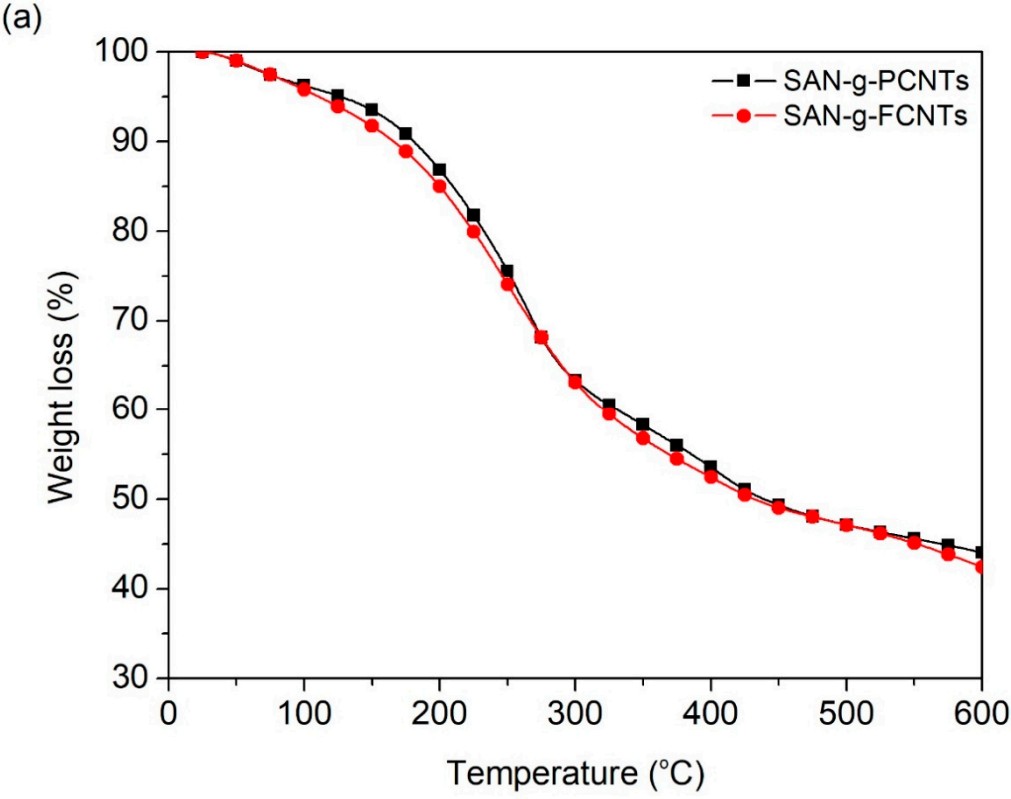

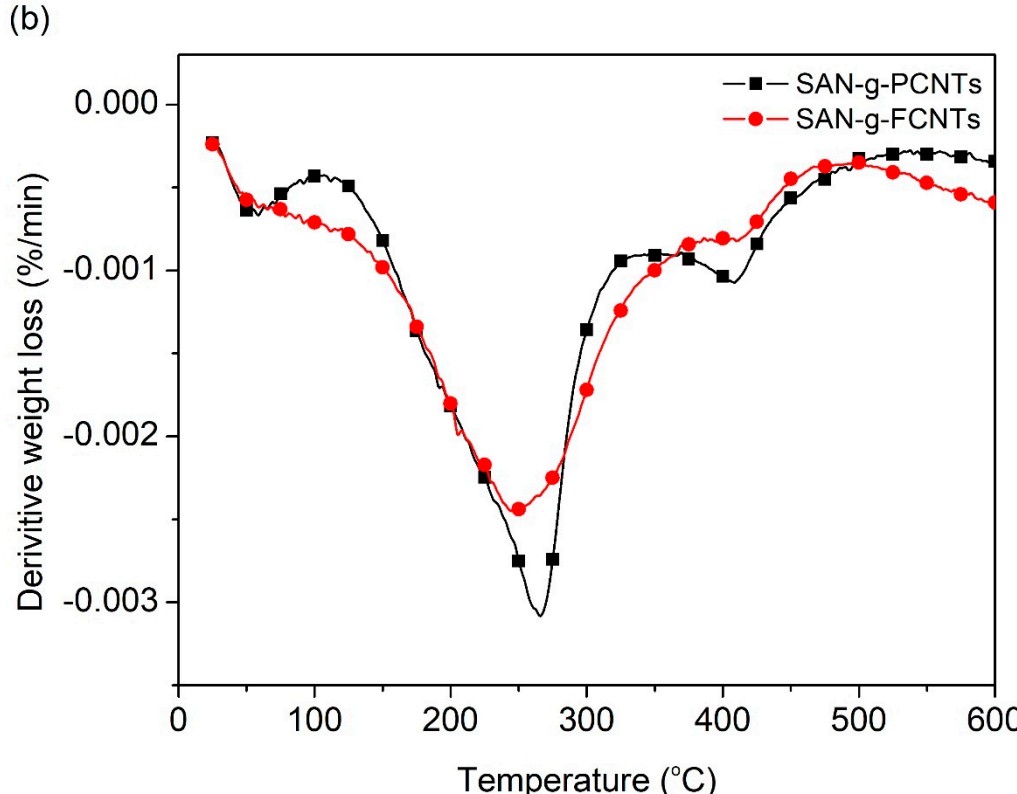

**Figure 8.** *Cont.*

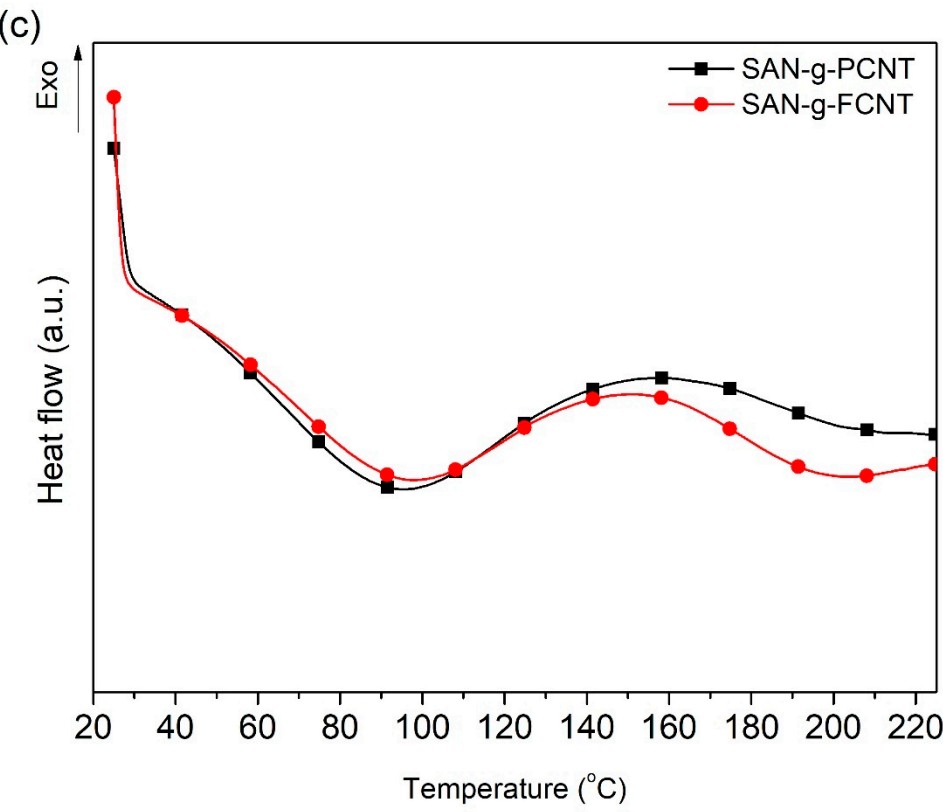

**Figure 8.** (**a**) TGA, (**b**) DTG, and (**c**) DSC curves for SAN-g-PCNT and SAN-g-FCNT.

DSC was further employed to study the effect of pre-functionalization of CNTs on the thermal behavior of the products fabricated by ATRP grafting of SAN onto MWCNTs. A typical result of a heating cycle on SAN-g-PCNT and SAN-g-FCNT is shown in Figure 8c. The SAN-g-PCNT and SAN-g-FCNT showed an endothermic peak at 102 and 105 °C, respectively. Such step changes are indicative of the occurrence of glass to rubber transition in the behavior of the polymer phase (SAN). These peaks can be designated as α relaxation, which is related to the glass-to-rubber transition temperature ($T_g$) of the soft segments [48]. It should be noted that these $T_g$ values were above the measured values of either PS or PAN pure components reported in the literature [49]. An increase in $T_g$ of the grafted copolymer in comparison with the pure component can be related to the tethering due to the interchain attractions and the restriction in the polymer chain mobility [43].

The figure reveals that the position of the peak temperature is affected by the surface characteristics of MWCNTs before ATRP, which subsequently controls the composition of SAN formed on the surface of MWCNTs by ATRP. It can also be seen that the functionalization of MWCNTs before SAN polymerization shifted the $T_g$ peak towards higher temperatures. The shift in the $T_g$ of SAN in SAN-g-FCNT was mainly attributed to the rise in the acrylonitrile content of SAN copolymer-grafted onto MWCNT. This can confirm that more acrylonitrile content in SAN copolymer could be achieved by using the FCNTs, as shown in the TGA and $^1$H-NMR analysis. Due to the polar nature of acrylonitrile units, the higher amounts of acrylonitrile in SAN phase increases the polarity of the polymers, which resulted in slightly higher $T_g$ values [48].

Figure 9 shows SEM images obtained from the morphology of SAN-g-PCNT and SAN-g-FCNT at various magnifications. These images revealed densely entangled networks of MWCNTs for both SAN-g-PCNT and SAN-g-FCNT. However, the surface of SAN-g-FCNT exhibits rougher surface morphology as compared with SAN-g-PCNT. The increased roughness of the surface of SAN-g-FCNTs indicated that the SAN-g-FCNTs were coated with a higher amount of polymer chains. This is further confirmed quantitively

by measuring the diameters of the SAN-grafted MWCNTs using image analysis software. Histograms of diameter distribution of SAN-grafted CNTs are shown in Figure 10.

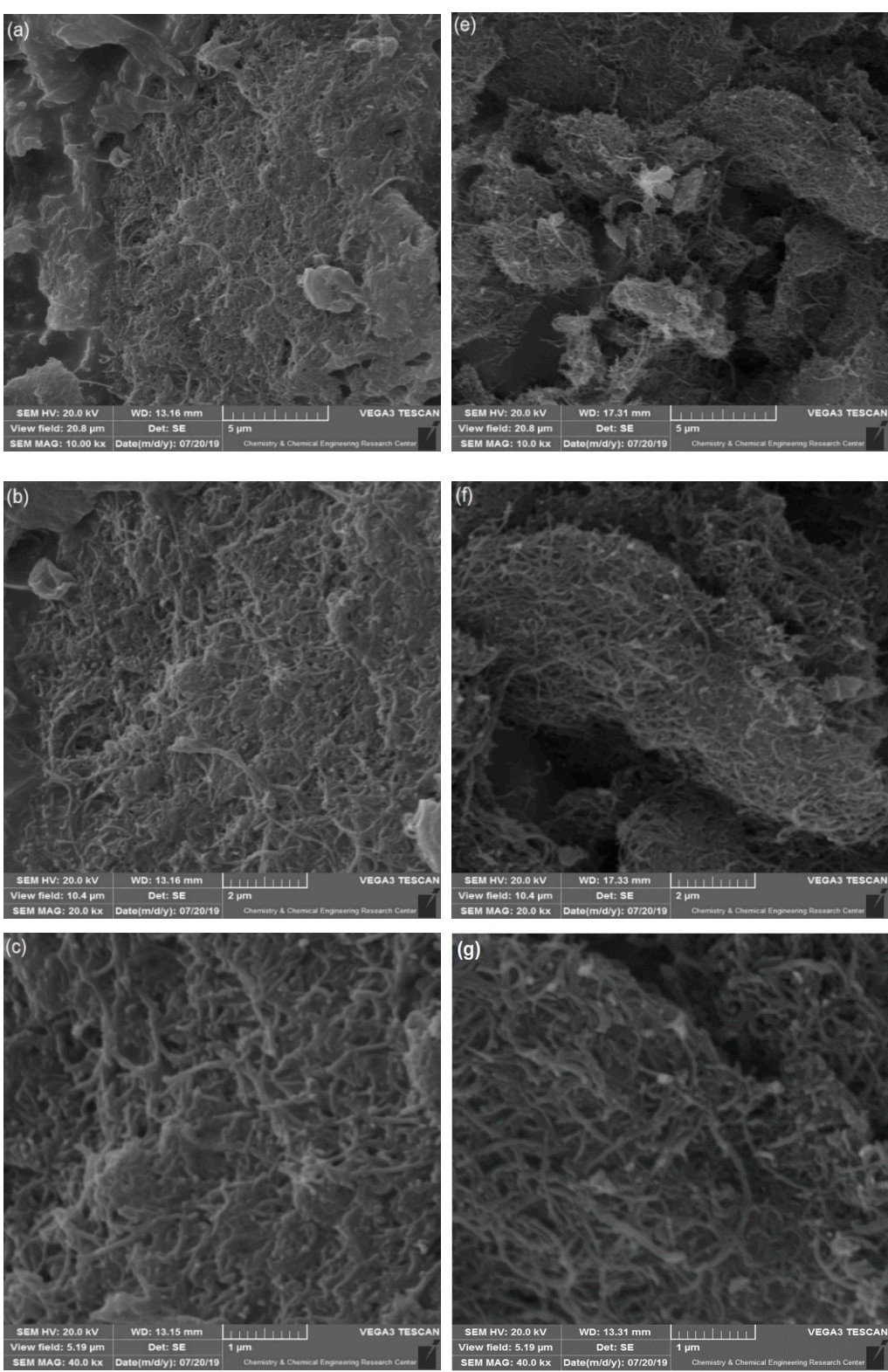

**Figure 9.** Scanning electron microscopy (SEM) micrographs of (**a**–**c**) SAN-g-PCNTs and (**e**–**g**) SAN-g-FCNTs.

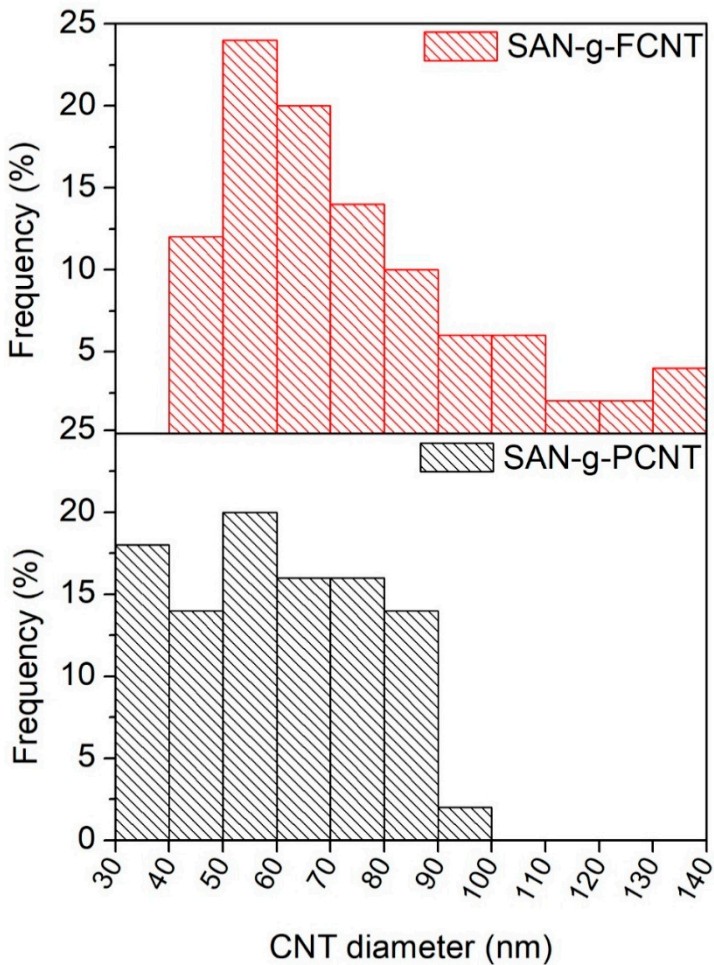

**Figure 10.** Diameter distribution histograms of SAN-grafted CNTs.

The external diameters of SAN-grafted MWCNTs exhibited a substantial increase when compared with the original MWCNTs, which provided further validation that there is SAN copolymer grafted onto the convex of MWCNTs through ATRP. It was evident from Figure 10 that acid functionalization of CNTs before ATRP of SAN significantly increased the diameters of the fabricated MWCNTs, indicating that the SAN-g-FCNTs were coated with a thicker layer of polymer chains. As expected, increasing the grafted polymer content onto MWCNTs raised their diameter. Overall, the SEM results provided further evidence of the successful grafting of SAN onto MWCNTs at higher polymer content using acid-functionalized MWCNTs, implying that pre-functionalization of CNTs display good SAN productivity in the ATRP method, and can be used as a suitable precursor for fabrication of SAN-grafted CNTs.

## 4. Conclusions

In this work, a vapor phase acid functionalization process was developed as a promising strategy for the preparation of high polymer content SAN-grafted CNTs via ATRP. The successful grafting of SAN polymer on the surface of the CNT was validated by using several characterization techniques, including FTIR, TGA, and [1]H-NMR. The results confirmed that SAN content grafted onto CNTs through the cyano group of acrylonitrile was significantly enhanced by a combination of vapor phase acid functionalization and surface-initiated ATRP. In other words, it was found that CNT-COOH is more suitable for macromolecule bonding. Furthermore, by the functionalized CNTs, acrylonitrile could be efficiently and selectively enriched in the SAN phase. The SEM results showed that

the functionalization of CNTs altered the surface roughness of the nanotubes. Moreover, a larger diameter of SAN-grafted functionalized CNTs was obtained as a result of the formation of a thicker layer of polymer macromolecules onto the surface of CNTs. This highly selective behavior of vapor-phase acid-functionalized CNTs opens possibilities to use these materials for surface-initiated ATRP-modified materials with improved polymer content for functional nanomaterials, nanocomposites, and other material applications.

**Author Contributions:** Conceptualization, E.E., A.V., and M.S.-K.; methodology, E.E. and M.S.-K.; software, M.A.; validation, E.E., A.V., and M.S.-K.; formal analysis, M.A.; investigation, M.A.; resources, E.E. and M.S.-K.; data curation, M.A.; writing—original draft preparation, M.A.; writing—review and editing, E.E., A.V., and T.H.M.; visualization, M.A. and E.E.; supervision, E.E. and M.S.-K.; project administration, E.E. and M.S.-K.; funding acquisition, E.E. and M.S.-K. All authors have read and agreed to the published version of the manuscript.

**Funding:** This research received no external funding.

**Institutional Review Board Statement:** Not applicable.

**Informed Consent Statement:** Not applicable.

**Data Availability Statement:** Not applicable.

**Conflicts of Interest:** The authors declare no conflict of interest.

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
