# Peer review of "Vapor Phase Modification for Selective Enrichment of Grafted Styrene/Acrylonitrile onto Carbon Nanotubes Via ATRP"

_processes, doi:10.3390/pr9030459_

Round 1
Reviewer 1 Report
The manuscript entitled, ‘Vapor phase modification for selective enrichment of grafted 2 styrene/acrylonitrile onto carbon nanotubes via ATRP’ reported synthesis of copolymer grafted CNTs. This work is good but there are several loopholes that should be countered before publication. I am recommending some concerns for this work.
- First of all the manuscript has lots of language problems, grammatical errors, tedious writing and very inadequacy in discussions. Those should be clarified with a sooth English writing.
- How did the author infer that the CNTs have been oxidized? There are no such proof in the manuscript.
- The TGA graph has no significant changes. Then why it is put so? It can be removed also.
- As per the DSC, it is recommended to discuss about their changes on glass transition temperature.
- The TEM images are very poor. I am recommending better and clear TEM images for those materials.
- The aim of the articles is not clear. The aim in introduction last paragraph should be modified.
- The ‘ml’ unit should be written as ‘mL’.
- The subheading ‘2.3 Characterization’ should be ‘2.2.3 Characterizations’.
- The unit in ‘mmoles/g’ is not the correct representation. It should be ‘mmole/g’.
- The surface behaviors of CNTs are not discussed. I am recommending some articles to refer for betterment of the discussions; 1016/j.nanoso.2020.100429; 10.1007/s10570-017-1441-4; 10.1016/j.synthmet.2021.116720; 10.1016/j.compositesa.2018.01.025.
Author Response
The manuscript entitled, ‘Vapor phase modification for selective enrichment of grafted 2 styrene/acrylonitrile onto carbon nanotubes via ATRP’ reported synthesis of copolymer grafted CNTs. This work is good but there are several loopholes that should be countered before publication. I am recommending some concerns for this work.
Authors’ response: The authors would like to thank the reviewer for the valuable comments which helped us to improve the quality of our manuscript.
- First of all the manuscript has lots of language problems, grammatical errors, tedious writing and very inadequacy in discussions. Those should be clarified with a sooth English writing.
Authors’ response: Thank you for the comment. Now, the whole document is revised and proofread, and language related and other corrections are made to improve the quality of the manuscript which can be tracked by the red color throughout the manuscript.
- How did the author infer that the CNTs have been oxidized? There are no such proof in the manuscript.
Authors’ response: Thanks to the reviewer’s comment, the formation of oxidized CNTs were confirmed by the higher carboxylic group concentration that was evaluated by Bohem titration, the disordered/rough surface of CNTs (decrease in graphitic structure) illustrated by TEM, relative lower structural crystallinity/disorder degree by Raman spectra, and higher weight loss of TGA curve.
- The TGA graph has no significant changes. Then why it is put so? It can be removed also.
Author’s response: The authors strongly believe that although the change in the curve is slight, it is still visible. It provides evidence that the oxidation took place and supports the idea that pre-oxidation of CNTs is a promising approach for the preparation of high polymer content SAN-grafted CNTs via ATRP where acrylonitrile could be efficiently and selectively enriched in the SAN. The results displayed by the TGA are in agreement with the other results and support the claims of the study. Therefore, the authors are convinced that there is value in keeping the TGA/DTG curves to support the discussion.
- As per the DSC, it is recommended to discuss about their changes on glass transition temperature.
Authors’ response: Thank you for the suggestion. In the discussion section of the revised manuscript, the authors have added extra explanation related to DSC as per the comment here.
- The TEM images are very poor. I am recommending better and clear TEM images for those materials.
Authors’ response: The authors agree that the high-quality TEM image would be more eye-catching but unfortunately HR-TEM was unavailable to us at this time. On the other hand, since we would like to see the exterior wall of CNTs to see the change in the morphology employing high magnifications was inevitable. Higher magnifications led to the significant drop in the quality of the images.
- The aim of the articles is not clear. The aim in introduction last paragraph should be modified.
Authors’ response: Thank you for the great comment. The authors revised the last paragraph of the introduction to highlight the novelty and aim of the work.
- The ‘ml’ unit should be written as ‘mL’.
Authors’ response: Thank you for the comment here. This is corrected throughout the manuscript.
- The subheading ‘2.3 Characterization’ should be ‘2.2.3 Characterizations’.
Authors’ response: It is corrected according to the reviewer’s comment.
- The unit in ‘mmoles/g’ is not the correct representation. It should be ‘mmole/g’.
Authors’ response: Thanks to the reviewer. It is corrected accordingly.
- The surface behaviors of CNTs are not discussed. I am recommending some articles to refer for betterment of the discussions; 1016/j.nanoso.2020.100429; 10.1007/s10570-017-1441-4; 10.1016/j.synthmet.2021.116720; 10.1016/j.compositesa.2018.01.025.
Authors’ response: Thanks to the reviewer’s invaluable comment, the discussion about the surface of the CNTs after functionalization and ATRP is now added to the text. The articles suggested by the reviewer are now added into the manuscript according to the style of the journal. The changes can be followed by red color.

Reviewer 2 Report
The manuscript describes a procedure to improving a vapor phase acid functionalization as a promising strategy for the preparation of high polymer content SAN-grafted CNTs via ATRP. The successful grafting of SAN polymer on the surface of the CNT was validated by using several characterization techniques including FTIR, TGA, and 1H-NMR. The results confirmed that SAN content grafted onto CNTs through the cyano group of acrylonitrile was significantly enhanced by a combination of vapor phase acid functionalization and sur- face-initiated ATRP. Furthermore, by the functionalized CNTs, acrylonitrile could be efficiently and selectively enriched in the SAN phase. The SEM results showed that the functionalization of CNTs altered the surface roughness of the nanotubes. Moreover, a larger diameter of SAN-grafted functionalized CNTs was obtained as a result of the formation of a thicker layer of polymer macromolecules onto the surface of CNTs. The results are well organized and presented tanks at a clear discussion and graphical representation. In addition, the manuscript is supported by an important review of the literature. For all these considerations, this manuscript could be accepted in present form.
Author Response
The manuscript describes a procedure to improving a vapor phase acid functionalization as a promising strategy for the preparation of high polymer content SAN-grafted CNTs via ATRP. The successful grafting of SAN polymer on the surface of the CNT was validated by using several characterization techniques including FTIR, TGA, and 1H-NMR. The results confirmed that SAN content grafted onto CNTs through the cyano group of acrylonitrile was significantly enhanced by a combination of vapor phase acid functionalization and surface initiated ATRP. Furthermore, by the functionalized CNTs, acrylonitrile could be efficiently and selectively enriched in the SAN phase. The SEM results showed that the functionalization of CNTs altered the surface roughness of the nanotubes. Moreover, a larger diameter of SAN-grafted functionalized CNTs was obtained as a result of the formation of a thicker layer of polymer macromolecules onto the surface of CNTs. The results are well organized and presented tanks at a clear discussion and graphical representation. In addition, the manuscript is supported by an important review of the literature. For all these considerations, this manuscript could be accepted in present form.
Authors’ response: The authors are very grateful to hear reviewer’s positive and supportive overall opinion.

Reviewer 3 Report
This paper investigates the acid-treated CNTs modified with PAN.
The paper is very interesting. I recommend that this paper is published after minor revisions.
The authors show the structures of SAN-g-PCNT and SAN-g-FCNT in Figures 2 and 7. However, the structural formula needs to be modified. Also, the explanation of the polymerization mechanism is insufficient.
Please consider the following points:
1) The connection part with CNT and SAN (-COO-C*=N-C*H-CH2-) is incorrect. The bond order of the carbon* is 3. Since acrylonitrile (N≡C-CH=CH2), C and N are also reversed.
2) The structure of the copolymer -(CH(C≡N)-CH2)m-(CH(Ph)-CH2)n- should be described.
3) Why is the absorption peak of C≡N not observed in the IR spectra of SAN-g-PCNT and SAN-g-FCNT?
4) It is necessary to explain in detail the radical polymerization mechanism of PCNT and FCNT. (Can carboxylic acids be the starting species for this radical copolymerization?)
Author Response
This paper investigates the acid treated CNTs modified with PAN. The paper is very interesting. I recommend that this paper is published after minor revisions. The authors show the structures of SAN-g-PCNT and SAN-g-FCNT in Figures 2 and 7. However, the structural formula needs to be modified. Also, the explanation of the polymerization mechanism is insufficient.
The authors would like to express their gratitude for the constructive opinions of the reviewer in improving this manuscript.
Please consider the following points:
- The connection part with CNT and SAN (-COO-C*=N-C*H-CH2-) is incorrect. The bond order of the carbon* is 3. Since acrylonitrile (N≡C-CH=CH2), C and N are also reversed.
Authors’ response: The authors would like to profoundly appreciate the reviewer’s precise comment on the graphical illustration of the SAN-g-PCNTs and SAN-g-FCNTs. We have revised figure 2 and 7 accordingly.
- The structure of the copolymer -(CH(C≡N)-CH2)m-(CH(Ph)-CH2)n- should be described.
Authors’ response: Thanks to the reviewer’s comment. The corrected structure of the copolymers is now illustrated in the manuscript.
- Why is the absorption peak of C≡N not observed in the IR spectra of SAN-g-PCNT and SAN-g-FCNT?
Authors’ response: Thanks to the reviewer’s precise comment. The authors expected to see a weak peak around 2000-2200 cm-1 related to C≡N as well. The reason why we didn’t see this peak was the fact that the intensity of the peak was so low that can not be detected here. The same behavior was observed for SAN-g-CNTs in the literature. In the figure below from [Journal of polymer science: part A: Polymer Chemistry, 45, 460-470 (2007)], it can be clearly seen that the peak due to C≡N can not be detected in the spectra. For the comparison, PAN-g-CNT exhibits a strong peak at 2100 cm-1. Therefore, we concluded that FTIR did not give much information about the polymerization of styrene or acrylonitrile monomers on the CNT surface. To understand it better, 1H-NMR was performed.
FTIR spectra for the polymer-grafted CNTs [Journal of polymer science: part A: Polymer Chemistry, 45, 460-470 (2007)]
- It is necessary to explain in detail the radical polymerization mechanism of PCNT and FCNT. (Can carboxylic acids be the starting species for this radical copolymerization?)
Authors’ response: Thanks to reviewer’s great comment, the explanation of how oxidation can affect the mechanism of ATRP polymerization of SAN is now added into the manuscript which is highlighted in red color. The explanation on the difference between After vapor phase treatment of PCNTs with nitric acid, the MWCNTs were functionalized with carboxylic acid at the open ends and the defects of the side walls. The carboxylic acid groups can effectively involve the fixation of the ATRP initiator onto the MWCNTs’ walls and the polymerization [J Polym Sci Part A: Polym Chem 45: 460–470, 2007].

Round 2
Reviewer 1 Report
The author tried to countered all the questions. It can be published in it's present form.